# Evolution-guided mutagenesis of the cytoplasmic incompatibility proteins: Identifying CifA's complex functional repertoire and new essential regions in CifB

J. Dylan Shropshire[1,2]*, Mahip Kalra[1,2], Seth R. Bordenstein[1,2,3,4]*

1 Department of Biological Sciences, Vanderbilt University, Nashville, Tennessee, United States of America, 2 Vanderbilt Microbiome Initiative, Vanderbilt University, Nashville, Tennessee, United States of America, 3 Department of Pathology, Microbiology, and Immunology, Vanderbilt University, Nashville, Tennessee, United States of America, 4 Vanderbilt Institute for Infection, Immunology, and Inflammation, Vanderbilt University Medical Center, Nashville, Tennessee, United States of America

* shropxp@gmail.com (JDS); s.bordenstein@vanderbilt.edu (SRB)

**Data Availability Statement:** All data generated in this study are available in the supplement (S1 Data of this manuscript.

## Abstract

*Wolbachia* are the world's most common, maternally-inherited, arthropod endosymbionts. Their worldwide distribution is due, in part, to a selfish drive system termed cytoplasmic incompatibility (CI) that confers a relative fitness advantage to females that transmit *Wolbachia* to their offspring. CI results in embryonic death when infected males mate with uninfected females but not infected females. Under the Two-by-One genetic model of CI, males expressing the two phage WO proteins CifA and CifB cause CI, and females expressing CifA rescue CI. While each protein is predicted to harbor three functional domains, there is no knowledge on how sites across these Cif domains, rather than in any one particular domain, contribute to CI and rescue. Here, we use evolution-guided, substitution mutagenesis of conserved amino acids across the Cif proteins, coupled with transgenic expression in uninfected *Drosophila melanogaster*, to determine the functional impacts of conserved residues evolving mostly under purifying selection. We report that amino acids in CifA's N-terminal unannotated region and annotated catalase-related domain are important for both complete CI and rescue, whereas C-terminal residues in CifA's putative domain of unknown function are solely important for CI. Moreover, conserved CifB amino acids in the predicted nucleases, peptidase, and unannotated regions are essential for CI. Taken together, these findings indicate that (i) all CifA amino acids determined to be crucial in rescue are correspondingly crucial in CI, (ii) an additional set of CifA amino acids are uniquely important in CI, and (iii) CifB amino acids across the protein, rather than in one particular domain, are all crucial for CI. We discuss how these findings advance an expanded view of Cif protein evolution and function, inform the mechanistic and biochemical bases of Cif-induced CI/rescue, and continue to substantiate the Two-by-One genetic model of CI.

**Funding:** This work was supported by National Institutes of Health awards R01 AI132581 and R01 AI143725, National Science Foundation award IOS 1456778, and the Vanderbilt Microbiome Initiative to S.R.B., and a National Science Foundation Graduate Research Fellowship DGE-144519 to J.D. S. The funders had no role in study design, data collection and analysis, decision to publish, or preparation of the manuscript.

**Competing interests:** J.D.S. and S.R.B. are listed as inventors on a provisional patent relevant to this work. S.R.B. is a coinventor on two other pending patents related to controlling arthropods.

## Author summary

*Wolbachia* are maternally-transmitted, intracellular bacteria that occur in approximately half of arthropod species worldwide. They can spread rapidly though host populations via the cytoplasmic incompatibility (CI) drive system. CI causes embryonic death when infected males mate with uninfected females, but offspring of infected females are rescued. Two proteins, CifA and CifB, underlie the genetic basis of CI and rescue, but how amino acid sites across these proteins contribute to CI and/or rescue remain unknown. Here, we employed evolution-guided, substitution mutagenesis on conserved amino acids to understand their relative contributions to CI and rescue. The results of this study reveal a phenotypic complexity underlying the expression of these proteins and provide relevance to the biochemical and mechanistic bases of CI and rescue.

## Introduction

*Wolbachia* are maternally-inherited, intracellular α-Proteobacteria that occur in 40–65% of all arthropod species [1–4]. Primarily residing in the cells of reproductive tissues, *Wolbachia* commonly cause a selfish drive system, cytoplasmic incompatibility (CI), that yields a relative advantage to females that transmit *Wolbachia* and thus increases *Wolbachia*'s rate of spread through the matriline [5,6]. CI causes embryonic death when *Wolbachia*-infected males mate with uninfected females and is rescued when the female is infected with a compatible *Wolbachia* strain (Fig 1A) [7–9]. CI can act as a form of reproductive isolation between populations of different infection states [10–13]. Additionally, this drive system has brought *Wolbachia* to the forefront of vector control efforts to combat Zika and dengue viruses because *w*Mel *Wolbachia* from *Drosophila melanogaster* flies confer resistance to RNA arboviruses when transinfected into *Aedes* mosquitoes [14–20]. Notably, *w*Mel-induced CI is the focus of this study.

CI's microbial genetic basis can be described by the Two-by-One genetic model (Fig 1B) [21] where the two phage WO genes, *cifA* and *cifB*, cause CI when dually expressed in testes [21–23], and *cifA* rescues CI when singly expressed in ovaries [21,24,25]. The Cif proteins are divided into at least five phylogenetic clades, referred to as Types 1–5 [23,26,27]. To date, this genetic model has been well supported using Type 1 *cif* genes from *w*Mel [21,23,24] and is consistent with results with Type 1 and 4 genes from *w*Pip of *Culex pipiens* mosquitoes [25,28]. However, the mechanism underlying CifA;B-induced CI and CifA-induced rescue remains mostly unresolved and limited to *in vitro* assays and structural homology-based predictive annotations discussed below.

CifA$_{w\text{Mel}}$ is weakly predicted to encode a catalase-related domain (catalase-rel), a domain of unknown function 3243 (DUF), and a sterile-like transcription factor domain (STE) (Fig 1C) [27]. Catalase-rel domains are predicted to catalyze the degradation of reactive oxygen species (ROS) [29,30]. DUF has a distant homology to globin-like domains and Puf-family RNA-binding domains, which influence the stability of eukaryotic RNAs [31,32]. Finally, STE domains mediate transcriptional induction in yeast [33]. Structural homology predictions identify the STE and Puf-family RNA-binding domains across four and five phylogenetic Cif Types respectively, whereas the catalase-rel domain is only annotated in Type 1 [26,27,73].

Conversely, CifB$_{w\text{Mel}}$ harbors two putative PD-(D/E)XK-like nuclease domains (PDDEXK), and a ubiquitin-like-specific protease 1 domain (Ulp1) (Fig 1C) [22,23,27]. The Ulp1 domain is restricted to Type 1 and 5 CifB's and cleaves poly-ubiquitin chains *in vitro* [22,73]. Since mutation of the CifB Ulp1's catalytic motif ablates CifB's ability to contribute to CI when transgenically expressed in *D. melanogaster*, the Ulp1 domain has previously been described as the

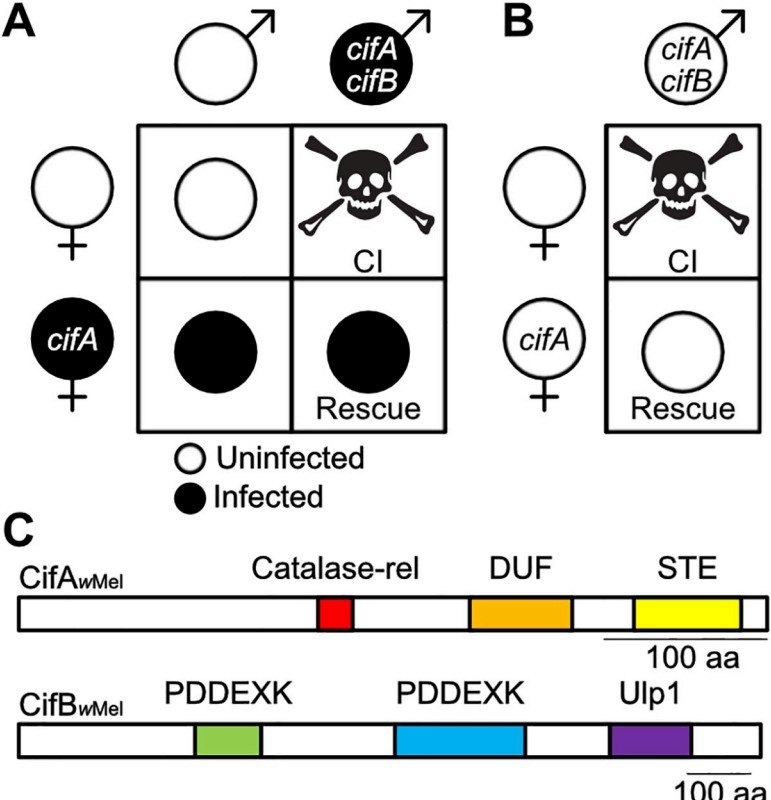

**Fig 1. Cytoplasmic incompatibility, the Two-by-One genetic model, and Cif protein architecture.** (A) CI is caused when *Wolbachia*-infected males expressing *cifA* and *cifB* mate with uninfected females. If females are infected and express *cifA* then offspring are rescued. (B) The Two-by-One genetic model indicates that males expressing *cifA* and *cifB*, even in the absence of an infection, can cause CI that can be rescued by uninfected females expressing *cifA* [21]. (C) Schematic showing the protein architecture of CifA and CifB from the *w*Mel *Wolbachia* of *D. melanogaster*. Annotations are based on a prior structural homology-based analyses [27].

"enzymatic warhead" for CI induction [22]. The function of the remaining sequence, including the PDDEXK dimer of Type 1 CifB, has not been assessed. We previously demonstrated that this remaining sequence comprises a CifB central core region found throughout Cif phylogenetic Types [23]. Moreover, the PDDEXK domains are also annotated in all phylogenetic Cif clades [23,26,27], and *in vitro* assays indicate that the PDDEXK domains of Type 4 CifB_{wPip} nick DNA [25]. As such, CifB's central core including the PDDEXK domains is likely important, if not one of the central aspects of its ability to contribute to CI. Finally, *in vitro* assays suggest that CifA and CifB can bind each other [22], but the importance of this binding *in vivo* and for phenotypic output remains unknown.

There are two debated models to explain Cif-induced CI and rescue [34–36]. The Host-Modification model posits that the Cif proteins interact and modify male products during spermatogenesis, resulting in CI when the modified sperm enter the embryo after fertilization [36]. Rescue occurs when maternally-derived CifA acts to reverse or otherwise inhibit the effect of the paternal modification. Conversely, the Toxin-Antidote model suggests that CifB acts as a toxin that is transported via the sperm to the embryo [34]. If CifA is not available to bind CifB and inhibit its toxicity, then the embryo will die due to CI. Thus, there are two key differences between these models. First, the Toxin-Antidote model requires that CifB is packaged in the sperm and transported to the embryo to cause CI, whereas the Host-Modification

model only requires the transfer of the sperm modification to the embryo [34,36]. Second, the Toxin-Antidote model requires that CifA acts as a discrete antidote in both CI and rescue by preventing CifB toxicity in spermatogenesis and embryogenesis respectively [34]. However, the Host-Modification model can explain instances where CifA has similar or different biochemical functions in CI and rescue [24,36]. Since it remains unknown if Cif products or the paternal modification travel with the sperm and how CifA contributes to CI and rescue, it cannot yet be concluded which of these models most accurately describes the CI phenotype [36].

The aforementioned protein annotations and biochemical data represent a foundation to develop a more complete and nuanced understanding of how Cif proteins cause CI and rescue, but it remains unclear what kind of impact each domain and unannotated region have on the phenotypic output of these proteins. Here, we test the importance of conserved amino acids across the Cif proteins to CI and rescue via site-directed substitution mutagenesis and transgenic expression in *D. melanogaster*. We report three key findings. First, conserved sites in CifA's N-terminal unannotated region and the catalase-rel domain are important in both CI and rescue. Second, conserved sites in CifA's DUF are only involved in CI. Finally, all tested conserved sites in CifB are required for CI. Taken together, we identify sites in seven Cif mutants (both CifA and CifB) essential for complete CI, and determine that CifA's N-terminus is involved in both CI and rescue while the middle of the protein is only involved in CI. These results inform the mechanistic and biochemical basis of CI and rescue and lend further support for a Two-by-One genetic model where both CifA and CifB are functionally crucial for expression of CI.

## Results

### CifA mutants impact CI and rescue

Since CifA does not have any putative catalytic motifs, we used a previous sequence analysis of conserved amino acid residues in an alignment of phylogenetically diverse CifA proteins [27] to select highly conserved sites across the protein for mutagenesis. $CifA_1$, $CifA_2$, $CifA_3$, and $CifA_4$ have combinatorial, alanine substitutions in the N-terminal unannotated region and putative catalase-rel, DUF, and STE domains, respectively (Fig 2A). Alanine mutagenesis is used to analyze the importance of specific amino acids in protein sequences without contributing significant structural variation to the protein [37]. We tested mutant *cifA* transgenes for their ability to (i) induce CI when dually expressed with wild-type *cifB* in testes of uninfected males and (ii) rescue when singly expressed in ovaries of uninfected females. Using the GAL4-UAS system in *D. melanogaster* [38], we predict that CI and/or rescue will not be recapitulated when CifA mutants are transgenically expressed if the sites mutated are crucial to the respective function. Notably, we use the *nos*-GAL4:VP16 driver throughout this study since it was previously shown to enable strong transgenic CI when dually expressing *cifA* and *cifB* transgenes [21]. Since CI manifests as embryonic death, we measured the strength of CI induced under mutant transgenic expression by measuring the percentage of *D. melanogaster* embryos that hatch into larvae.

Consistent with prior studies [21], dual *cifA;B* expression in males yielded nearly complete embryonic death typical of CI when mated to uninfected females (Mdn = 0% hatching) that was statistically comparable to *w*Mel-induced CI (p > 0.99), and it was rescued by *w*Mel-infected females (Mdn = 95.4% hatching) (Fig 2B). Transgenic dual expression of either *cifA_1; B* (Mdn = 93.7% hatching; p > 0.99) or *cifA_3;B* (Mdn = 94.1% hatching; p > 0.99) in males crossed to uninfected females revealed no statistically significant difference in hatching relative to the compatible, rescue cross; thus, mutating conserved sites in CifA's unannotated region and putative DUF ablates CI. Conversely, transgenic expression of *cifA_4;B* caused hatch rates

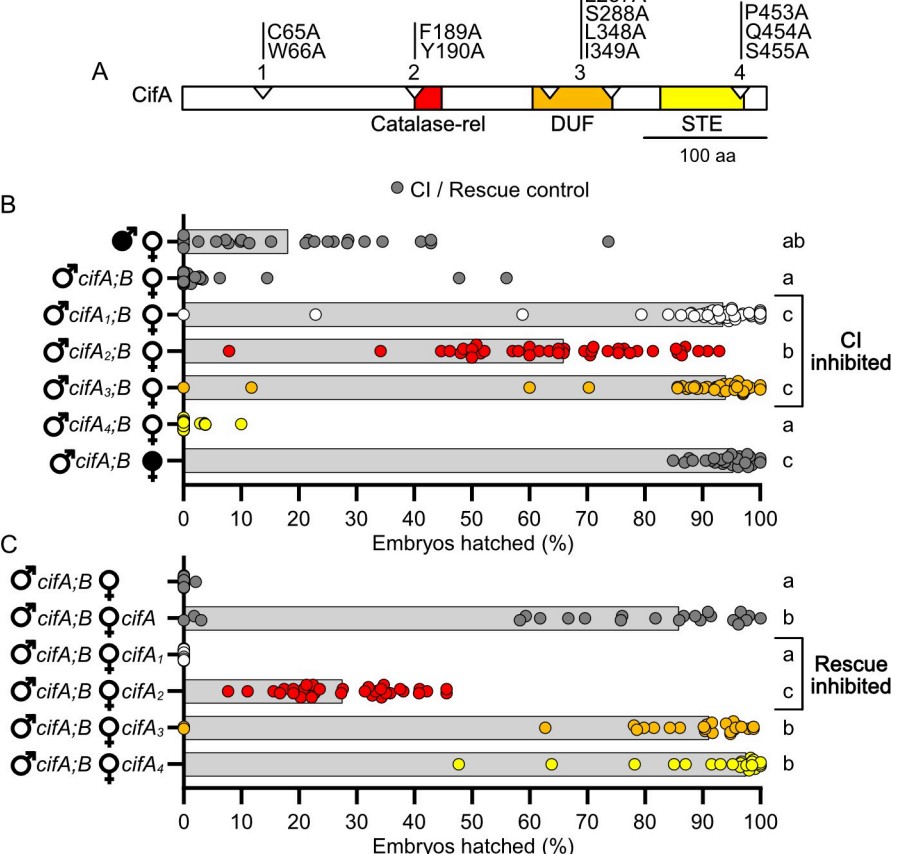

**Fig 2. *cifA₁* and *cifA₂* fail to cause or rescue CI, and *cifA₃* can rescue but fails to cause CI.** (A) schematic showing the location of amino acid mutations in CifA relative to previously predicted domains [27]. (B) Hatch rate experiment testing if *cifA* mutants can induce CI when dual expressed with *cifB* in uninfected males. (C) Hatch rate experiment testing if expressing *cifA* mutants can rescue transgenic CI when expressed in uninfected females. (B/C) Each dot represents the percent of embryos that hatched from a single male and female pair. Expressed genes are noted to the right of the corresponding sex. Gray bars represent median hatch rates for each cross and letters to the right indicate significant differences based on α = 0.05 calculated by Kruskal-Wallis and Dunn's test for multiple comparisons between all groups. Panel B was conducted three times and C was conducted twice. P-values are reported in S1 Table.

statistically comparable to *cifA;B*-induced CI (Mdn = 0% hatching; p > 0.99), suggesting that mutation of conserved sites in the putative STE did not impact *cifA*'s ability to contribute to CI. Finally, transgenic expression of *cifA₂;B* (Mdn = 66.0% hatching) yielded an intermediate phenotype whereby it was statistically different from both *cifA;B*-induced CI (p = 0.0006) and rescue of transgenic CI (p = 0.0001), indicating that the putative catalase-rel mutant induces a partial CI phenotype. Together, these results suggest the mutated sites in the unannotated region, catalase-rel, and DUF of CifA are important for CI-induction (Fig 2B).

Next, we reciprocally tested if uninfected transgenic females singly expressing the same *cifA* mutants rescue *cifA;B*-induced CI (Fig 2C) as previously established [21]. *cifA;B* expressing males were chosen over *w*Mel-infected males to precisely compare mutant and wild-type variants under a common transgenic expression system as well as to avoid confounding variables with *w*Mel biology such as male age, paternal grandmother age, and developmental timing effects on CI penetrance [21,39–41]. As above, dual *cifA;B* expressing males induced near-complete embryonic death consistent with strong CI (Mdn = 0% hatching), and this lethality could be rescued when the female expressed *cifA* (Mdn = 85.9% hatching). Transgenic

expression of $cifA_1$ (Mdn = 0.00% hatching; p < 0.0001) and $cifA_2$ (Mdn = 27.6% hatching; p = 0.0390), which failed to contribute to CI, also failed to rescue $cifA;B$-induced CI as compared to the standard transgenic rescue cross. Conversely, transgenic expression of $cifA_3$ (Mdn = 91.2% hatching; P > 0.9999) and $cifA_4$ (Mdn = 97.6% hatching; P = 0.3039) rescued $cifA;B$-induced CI at levels comparable to the standard transgenic rescue cross. These results suggest that the sites mutated in the unannotated and catalase-rel regions of CifA are important for rescue (Fig 2C).

## CifB mutants ablate CI

Four CifB mutants were constructed based on a comparative sequence analysis of conserved residues [27]. All CifB mutations are similarly alanine substitutions, with the exception of one glycine mutation of a conserved alanine (Fig 3A). Glycine was chosen to replace alanine since it is comparably sized and would be less likely to impact protein structure than other amino acids. $CifB_1$, $CifB_2$, $CifB_3$, and $CifB_4$ have mutations in the N-terminal unannotated region, first PDDEXK, second PDDEXK, and Ulp1 respectively (Fig 3A). The Ulp1 mutation is the same used previously to test for the catalytic activity of the Ulp1 domain [22]. We predict that CI will not be recapitulated when CifB mutants are transgenically expressed if the sites mutated are crucially important for CI-induction. As with CifA mutants above, we tested mutant CifB for their ability to induce CI when dually expressed with wild-type $cifA$ in uninfected males.

As expected, dual $cifA;B$ expression in uninfected males caused hatch rates statistically comparable to $w$Mel-induced CI (p > 0.99), and it could be rescued by $w$Mel-infected females (Mdn = 93.9% hatching). However, transgenic expression of $cifA;B_1$ (Mdn = 96.3%; p < 0.0001), $cifA;B_2$ (Mdn = 95.6%; p < 0.0001), $cifA;B_3$ (Mdn = 94.3%; p < 0.0001), and $cifA;B_4$ (Mdn = 93.0%; p < 0.0001) all failed to reduce hatch rates statistically comparable to $cifA;B$-

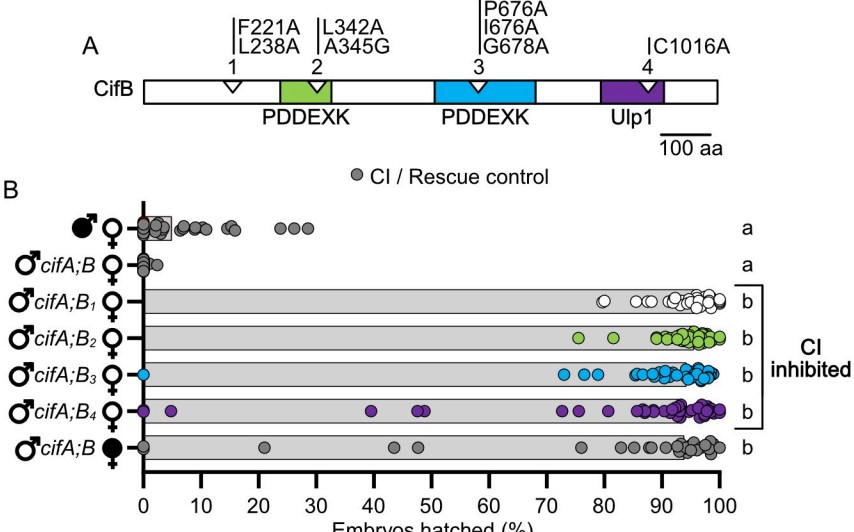

**Fig 3. All *cifB* mutants fail to contribute to CI.** (A) schematic showing the location of mutations in CifB relative to previously predicted domains [23,27]. (B) Hatch rate experiment testing if *cifB* mutants can induce CI when dual expressed with *cifA* in uninfected males. Each dot represents the percent of embryos that hatched from a single male and female pair. Expressed genes are noted to the right of the corresponding sex. Gray bars represent median hatch rates for each cross and letters to the right indicate significant differences based on α = 0.05 calculated by Kruskal-Wallis and Dunn's test for multiple comparisons between all groups. Panel B was conducted three times. P-values are reported in S1 Table.

induced CI (Mdn = 0.0%). These results specify that all mutated conserved sites, rather than any one site or domain such as the previously reported catalytic site of Ulp1 [22], are important for CifB in CI-induction.

## CifB selection analysis

Prior selection analyses using a sliding window analysis indicated that CifA is under strong purifying selection, and CifA's N-terminal unannotated region and catalase-rel domain are under stronger purifying selection than C-terminal regions [24]. Here, in order to assess how engineered mutations and phenotypes align with selective pressures, we replotted this analysis for CifA and, for the first time, applied a sliding window analysis of $K_a$ and $K_s$ (SWAKK) to assess the selective pressure on CifB (Fig 4). SWAKK calculates the ratio of non-synonymous to synonymous substitution rates ($K_a/K_s$) in a pairwise alignment. Codons with ratios below one are considered under purifying selection, and those above one are under positive selection. In both cases, pairwise nucleotide alignments were $cif_{wMel}$ relative to $cif_{wHa}$, which are modestly divergent Type 1 $cif$ variants. For both CifA and CifB, we then plotted the amino acid substitutions to assess if they occur in regions of purifying or positive selection.

For CifA, all mutant sites were in areas of purifying selection (Fig 4). However, the strongest purifying selection was surrounding the codons we mutated in the N-terminal unannotated region and catalase-rel domain. Intriguingly, the sites mutated near CifA's STE are under strong purifying selection, but the rest of the domain evolves neutrally. These results are in-line with our observations that the mutations in the N-terminal unannotated region and

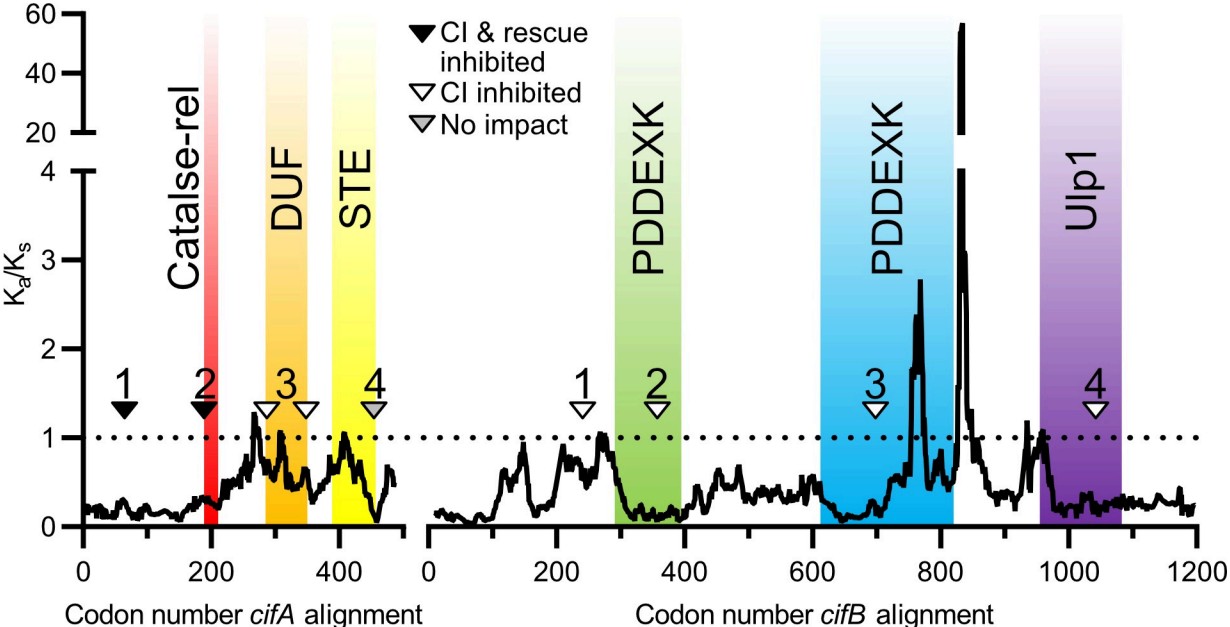

**Fig 4. Sliding window analyses reveal selective pressures on regions surrounding evolution-guided mutagenesis.** Analyses are based on pairwise alignments of $cifA$ and $cifB$ variants of $w$Mel and $w$Ha where the ratio of non-synonymous to synonymous substitutions ($K_a/K_s$) are calculated along a 25 amino acid sliding window. The analysis of CifA was previously reported [24]. The horizontal dotted line represents a $K_a/K_s$ value of 1, indicating neutrality. Residues with $K_a/K_s$ values above the dotted line are under positive selection and those under the line are under purifying selection. Both proteins are largely under strong purifying selection, but CifB has two regions of positive selection. Sites mutated in $CifA_1$, $CifA_2$, $CifA_4$, $CifB_2$, $CifB_3$, and $CifB_4$ are in regions of purifying selection. Sites mutated in $CifA_3$ and $CifB_1$ appear to be under neutrality or slight purifying selection. Triangles on protein schematics represent a selection of mutated amino acids. The specific amino acids mutated are further illustrated in Figs 2A and 3A for CifA and CifB respectively. Annotations are from prior studies and based on structural homology based analyses [27].

catalase-rel domain are involved in both CI and rescue, whereas weaker selection may act on the DUF domain which based on the results here is only involved in CI. Additionally, as with CifA, the majority of CifB is under strong purifying selection (Fig 4). However, there are two clusters of amino acids that appear to be under strong positive selection. These residues are on the C-terminal side of CifB's second PDDEXK domain and in the unannotated region between the PDDEXK domain and Ulp1 domain. $CifB_1$'s mutations fall within regions under neutrality, whereas the other CifB mutants are in regions under strong purifying selection.

## Cif structural predictions

There are numerous ways to interpret the impact of a mutation on a protein's function. These can include non-exclusive changes to catalytic motifs, ligand binding sites, or changes in local or global structures that ablate, enhance, or otherwise modify the phenotypic output of the protein. The $CifB_4$ mutation in the Ulp1 domain involves a putative cysteine catalytic motif common to deubiquitinase domains [22]. Otherwise, no other catalytic motifs or binding sites have been identified in $CifA_{wMel}$ or $CifB_{wMel}$. As such, we aimed to investigate the impact of these mutations on the structure of CifA and CifB proteins.

The Iterative Threading ASSEmbly Refinement (I-TASSER) webserver was used to generate a list of structural homologs from the protein databank (PDB) for each wild-type and Cif mutant protein and construct structural models based on these hits [42]. The shared and unique PDB hits for wild-type and mutant proteins are summarized in (S1A–S1C Fig and detailed in S2 Table. The top 10 PDB hits for each protein were used to create structural models (S1B and S1D Fig). Each model is generated with confidence measures in the form of C-scores and TM-scores. C-scores range from -5 to 2 where 2 is the highest confidence, and TM-scores range from 0–1 where 1 is the highest accuracy [42]. The similarity between wild-type and mutant structures was then assessed using the Alignment plugin in PyMOL 2.3.2 which provides values for the root-mean-square deviation (RMSD) of atomic positions [43]. Higher RMSDs indicates a greater distance between the atoms of mutant proteins superimposed on the wild-type protein. Structural models were generated for CifA (C-score = -2.74;TM = 0.4 ±0.13), $CifA_1$ (C-score = -1.42; TM = 0.54±0.15; RMSD = 19.7), $CifA_2$ (C-score = -2.84; TM = 0.39±0.13; RMSD = 1.9), $CifA_3$ (C-score = -1.39;TM = 0.54±0.15; RMSD = 20.2), and $CifA_4$ (C-score = -1.43;TM = 0.54±0.15; RMSD = 19.8) (S1B Fig). These low C-scores and TM-scores indicate that the I-TASSER predictions for CifA are not robust, and that variation in structure between wild-type and mutant proteins could be the result of poor threading templates. However, these results suggest that CifA is structurally most comparable to $CifA_2$ while the other structures are predicted to change to comparable degrees. Crucially, since the most divergent model, $CifA_3$, remains rescue-capable and the second most divergent model $CifA_4$ remains functional in both CI and rescue it is unlikely that predictions of global structural variation assists an understanding of phenotypic ablation in CifA.

I-TASSER was also used to identify PDB hits and create structures for wild-type and mutant CifB. The shared and unique PDB hits for wild-type and mutant proteins are summarized in S1C Fig and detailed in S2 Table. As above, I-TASSER protein structures were then created based on these threading templates and compared for RMSD. Structural models (S1D Fig) were generated for CifB (C-score = -1.02, TM-score = 0.59±14), $CifB_1$ (C-score = -0.68, TM-score = 0.63±14; RMSD = 1.7), $CifB_2$ (C-score = -1.03, TM-score = 0.58±14; RMSD = 10.6), $CifB_3$ (C-score = -1.07, TM-score = 0.58±14; RMSD = 2.2), $CifB_4$ (C-score = -0.65, TM-score = 0.63±14; RMSD = 2.2) (S1D Fig). Together, these results suggest that all CifB mutants are structurally comparable to the wild-type protein. As with CifA, it remains unknown how small effects in protein structure may influence phenotypic ablation.

## Discussion

CI's genetic basis involves a Two-by-One genetic model whereby *cifA;B* expressing males cause CI and *cifA* expressing females rescue CI [21–24,28]. However, the mechanistic basis of *cifA;B*-induced CI and *cifA*-induced rescue remains largely unresolved. Here, we test the phenotypic impact of conserved sites [27] to CI and rescue across the Cif_wMel proteins *in vivo* using site-directed mutagenesis and transgenic expression in *D. melanogaster*. We discuss the relevance of these findings, particularly the complex functional repertoire of CifA, to the Toxin-Antidote and Host-Modification models of CI [34,36] and the molecular basis of CifA; B-induced CI and CifA-induced rescue.

Crucially, phenotypic ablation caused by site-directed substitution mutagenesis has multiple possible and non-exclusive interpretations. First, a mutated site may be part of a catalytic motif. If so, mutagenesis may result in a significant inhibition of enzymatic activity related to CI and/ or rescue-induction. Second, the mutated site may be crucial for binding to host ligands, binding between CifA and CifB, or binding to nucleotide products. Additionally, amino acid substitutions may impact protein structure which in turn could also ablate function. Global folding abnormalities may block the enzymatic function of trans-acting domains or prevent binding elsewhere in the protein. Local or non-local changes to the structure can inhibit processes close to the mutated site in tertiary structure [44,45]. These hypotheses are applicable to all mutant variants. Below, we place our mutagenesis work in the context of prior *in vivo* assays and bioinformatics to develop hypotheses regarding the mechanistic basis of CI and rescue.

First and foremost, our results indicate that CifA has several amino acids with overlapping functions in CI and rescue, as well as amino acids of specific importance to CI. Thus, it is plausible that, CifA's functional role in CI is comparable to that in rescue. Among the two sets of mechanistic models for CI, Host-Modification-based models predict that expression of CifA and CifB cause CI through host modifications during spermatogenesis, and rescue occurs by reversing those modifications in the embryo [36]. In the context of these results, CifA's role in modifying host factors in the testes may be comparable to its activity in rescue. Indeed, the Host-Modification-based mistiming model posits that CI causes a delay in nuclear apposition during the first mitosis due to slowed development of the male pronucleus, and rescue occurs when the development of the female pronucleus is comparably delayed [46,47]. Thus, under the mistiming model, CifA's overlapping function in CI and rescue could indicate that it is the primary driver of CI-induction and rescue, whereas CifB and CI-specific sites in CifA may provide adjunct functions necessary to access those targets when expressed in the testes. Conversely, the Toxin-Antidote model predicts that CifB is paternally-transferred and the primary factor that causes an embryonic toxicity, and CifA is necessary to prevent self-induced toxicity in the testes and to rescue CifB-toxicity in the embryo [34]. Unlike the Host-Modification model, the Toxin-Antidote model predicts that CifA must function exactly as an antidote when expressed in testes and embryos. Thus, our data are also in-line with these expectations, but as with the Host-Modification model, the Toxin-Antidote model must be expanded to explain why sites in CifA's DUF domain are only essential for CI-induction if CifA's primary function is as an antidote in both tissues. Crucially, there is no evidence in the literature that CifB is transferred to the embryo, and we have proposed a Host-Modification model wherein CifA is both a primary inducer of CI in testes and rescue in embryos, while CifB is an accessory protein; this scenario enables a simple, one-step mutation scenario to the evolution of bidirectional CI between different strains of *Wolbachia* [24]. More work will be necessary to understand CifA's cell biology and biochemistry to support it.

In the context of CifA, mutations in the N-terminal region completely ablate both CI and rescue. Intriguingly, prior selection analyses revealed that this region is under stronger

purifying selection than the C-terminal end of the protein, suggesting that conservation in CifA's N-terminus may have a strong impact on its phenotypic output [24]. Given that these N-terminal regions in both CifA and CifB are unannotated, we cannot yet provide a specific explanation for why these regions ablate CI and rescue. However, it is notable that N-terminal regions in *w*Pip homologs of both CifA and CifB are predicted to encode ankyrin-interacting domains [48]. Thus, these mutations may be responsible for ablating the binding capacity of Cif proteins to host ligands through either sequence or structural modifications of the protein, which requires further study.

When conserved sites within CifA's putative catalase-rel domain were mutated and dually expressed with CifB in males, 65.95% of embryos hatched indicating that CI capability was inhibited but not ablated. Similarly, when expressed in females crossed to CI-inducing CifA;B males, only 35.45% of embryos hatched, indicating that rescue was also significantly weaker. Thus, these amino acids are important for causing complete CI and rescue phenotypes. Catalases are enzymes that are involved in the decomposition of hydrogen peroxide and protect cells from reactive oxygen species (ROS) damage [30]. Some catalase-like domains are involved in host immune pathways that use ROS to combat disease [49,50], and high levels of ROS can cause male infertility in organisms as diverse as *Drosophila* and humans [51,52]. Notably, while CifA is annotated with a catalase-rel domain [27], the closest sequence homolog is from *Helicobacter pylori*, which shares only ~22% sequence identity [29] and has no obvious active sites [27]. Thus, it is unknown if CifA's catalase-rel is capable of degrading ROS, but it may otherwise attract or interact with ROS while not degrading them. For example, oxidative posttranslational modifications (PTM) can shift phenotypic output [53]. CifA's catalase-rel may attract ROS, enabling PTM of itself, CifB, or other host targets. Since oxidative PTMs can be reversible [53], rescue may in part occur through the removal of these PTMs in the embryo. Alternatively, CifA may help to localize ROS to host targets to induce oxidative damage or otherwise modify host targets.

Moreover, *Wolbachia* presence is correlated with increases in ROS in *D. melanogaster*, *D. simulans*, *A. albopictus*, *A. polynesiensis*, and *T. urticae* [50,54,55], yielding additional support for CifA's function as a catalase-related protein [27]. *Wolbachia*-induced increases in ROS levels correlate with DNA damage in *D. simulans* spermatocytes [54] and an increase in lipid hydroperoxides in *D. melanogaster*, which are markers for ROS-induced oxidative damage [56]. Intriguingly, the immune-related gene *kenny* (*key*) is upregulated in *Wolbachia*-infected *D. melanogaster*, and experimental upregulation of *key* in uninfected male flies yielded increased ROS levels, DNA damage, and decreased hatching that can be rescued when mated to infected females [57]. Together, these data support a role for ROS in CI's mechanism, but more work is necessary to link CifA's catalase-rel domain to ROS variation and determine if ROS are directly responsible for CI/rescue-induction or are otherwise a symptom of other modifications in gametogenesis.

The DUF in CifA shares distant homology to Puf-family RNA-binding proteins and is the only putative domain shared in all five CifA clades [26]. Notably, mutating conserved residues in CifA's DUF domain revealed their specific importance to CI-induction but not for rescue. As such, this was the only domain in CifA wherein mutated sites were differentially important between the two phenotypes. RNA-binding proteins are important in transcriptional regulation and can influence the stability, localization, and translation of bound RNA. Puf-family RNA-binding proteins typically influence the stability of mRNAs involved in cell maintenance, embryonic development, and other processes [58–60]. For example, the *Drosophila* Puf-family RNA *Pumilio* (*pum*) is crucially involved in the establishment of patterning and abdominal segmentation in early embryonic development by suppressing the translation of maternal hunchback RNA in the *Drosophila* embryo [60,61]. Moreover, *pum* in spermatogenesis

negatively regulates the expression of p53 which is involved in DNA repair and apoptosis, increases apoptosis, and reduces sperm production and fertility [62]. Intriguingly, mitochondrial protein p32 is a candidate suppressor of CI based on *in vitro* pull-down assays using *Drosophila* lysates, and p32 regulates p53 activation [63], suggesting that *pum* in spermatogenesis may influence similar pathways in CI. Additionally, *Wolbachia* infection has been shown to have a considerable impact on the fly transcriptome including sRNA profiles [64–66]. On their own, these correlations do not sufficiently link transcription with CI. However, as described above, *key* is significantly upregulated and causes rescuable hatch rate defects when experimentally overexpressed [57]. Additionally, *Wolbachia* upregulate the sRNA nov-miR-12 which negatively regulates *pipsqueak* (*psq*), a DNA-binding protein that impacts chromatin structure [67,68], and knockdown of *psq* causes CI-like embryonic abnormalities and hatch rates in *D. melanogaster* [66]. Thus, CifA's DUF may influence the expression of RNAs involved in the CI pathway, and by mutating the conserved site it may ablate the domain's ability to regulate these RNAs. It is also important to note that since the DUF mutant only prevented CifA from contributing to CI, it supports prior hypotheses that CifA has distinct mechanistic input to CI and rescue [21]. The phenotypic plasticity of CifA may be caused by distinct protein conformations in testes and ovaries or DUF-associated targets may only be present in testes. More work will be necessary to confirm that CifA can bind RNAs and what impact this binding has on downstream processes.

The final CifA domain shares homology to STE transcription factor proteins which are found predominantly in fungi and encode a sequence-specific DNA-binding motif that influences yeast reproduction through pheromone-responsive elements [33]. Mutation of conserved sites within the STE had no impact on either CI or rescue. This was surprising since the STE domain appeared structurally conserved across four of the five CifA phylogenetic Types [26,27]. However, while the sites mutated in the domain were conserved, the remainder of the domain appears to be evolving mostly neutrally [24], suggesting the domain may be of lesser importance than N-terminal regions. Alternatively, since transgenes are expressed via host transcription and translation machinery, transgenic expression bypasses the need to export the proteins outside of *Wolbachia* and through the host-derived membranes that surround *Wolbachia* within the cell [69,70]. As such, it is possible that the STE domain has an essential functional role in translation initiation and/or in protein export within *Wolbachia*, which may or may not be related to CI and/or rescue-induction. Additional research will be necessary to determine if any component of the STE domain is necessary for CI and whether this domain is essential when expressed inside *Wolbachia*.

Type 1 CifB have two domains with homologs in the PDDEXK nuclease family, but they do not encode a canonical PD-(D/E)XK catalytic motif [22,71]. This nuclease family is heavily involved in DNA restriction, repair, recombination, and binding [71]. Mutations in conserved sites in either domain of Type 1 CifB$_{wMel}$ ablated CI phenotypes. Interestingly, homologs of CifB proteins from all five phylogenetic Types harbor putative nuclease domains [27], and the Type 4 CifB have functional nucleases with canonical catalytic motifs and can induce CI upon dual expression with CifA [25]. However, it remains biochemically unclear how these DNA nicks contribute to wild-type CI induction, and how this can be rescued by CifA expressing females. Moreover, while we show here that mutating conserved residues in either PDDEXK domain ablates CI induction, this does not confirm its role as a nuclease. For instance, these domains may be essential for the localization of Cif proteins to host DNA or other host targets. More work will be necessary to determine if CifB's PDDEXK domains are indeed active nucleases and why mutating these conserved sites ablates CI. Though, we note that ablation of CI by site-specific mutagenesis across the Cif proteins highlights the utility of a mechanistically-agnostic, gene nomenclature, like the CI factor (*cif*) gene nomenclature, as it is increasingly

clear that CifB's role as a deubiquitinase or nuclease alone does not define its role in CI [34,36].

CifB's final domain is a Ulp1 domain that contains the only known catalytic motif within the Cif proteins and is responsible for the deubiquitinase activity observed *in vitro* [22]. Previous reports show that mutating the conserved cysteine active site ablates CI function in CifB$_{w\text{Pip}}$ [22]. Here, we confirm that mutating the same cysteine active site ablated CI in the CifB$_{w\text{Mel}}$ protein. However, it is premature to claim that the Ulp1 domain is a "catalytic warhead" for CI [22] because several sites, when mutated, ablate the CI phenotype. Instead, it is evident that this site in the Ulp1 domain plays an important role in CI-induction but, when analyzed in context of results from mutating other regions, it is not the only crucially important component of the protein. More work will be necessary to dissect the relative importance of the unannotated region, the nuclease domains, and the Ulp1 in CI's mechanism.

In conclusion, we report conserved amino acids in CifA and CifB that are essential for CI and rescue phenotypes. For CifA, conserved sites in the unannotated region and catalase-rel domain were important for CifA-induced CI and rescue, while the mutated sites in the DUF was specifically important to CI. For CifB, mutating conserved sites in an unannotated region, both PDDEXK nuclease domains and the Ulp1 domain were important in CifB-induced CI. These works provide additional support for the necessity of expressing both CifA and CifB proteins to cause CI and the importance of CifA's complex functional repertoire and new essential regions in CifB to the genotype-phenotype relationship and mechanism underpinning CI.

## Materials and methods

### Creating transgenic flies

*cifA$_1$* and *cifB$_1$* mutant transgene variants were synthesized *de novo* at GenScript and cloned into a pUC57 plasmid. Site-directed mutagenesis was then performed by GenScript to produce the remaining three mutant variants of each gene (Figs 2A and 3A). UAS transgenic *cifA* and *cifB* mutant flies were then generated following previously described protocols [23]. Each gene was subcloned into the pTIGER plasmid through GenScript, which is a pUASp-based vector designed for germline expression. *cifA* and *cifB* transgenes were integrated into the y$^1$ M{vas-int.Dm}ZH-2A w$^*$; P{CaryP}attP40 and y$^1$ w$^{67c23}$; P{CaryP}attP2 attachment sites in the *D. melanogaster* genome respectively using PhiC31 integrase via embryonic injections at Best-Gene. At least 200 *D. melanogaster* embryos were injected per construct. Successful transformants were screened based on a red eye color marker included on the pTIGER plasmid. All wild-type and transgenic nucleotide and amino acid sequences are reported in S3 Table.

### Fly rearing and strains

*D. melanogaster* stocks *y$^1$w$^*$* (BDSC 1495), *nos*-GAL4:VP16 (BDSC 4937), and UAS transgenic lines homozygous for *cifA*, *cifA* mutants, *cifB*, *cifB* mutants, *cifA;B*, and lines dual homozygous for *cifA* or *cifB* mutants with wild-type counterparts were maintained on a 12-hour light/dark cycle at 25°C on 50mL of standard media. Dual transgenic lines were generated through standard genetic crossings and were all homozygous viable. Uninfected lines were produced by three generations of tetracycline treatment (20 μg/ml in 50 ml of fly media) for use in previous studies [23]. Infection status for all lines was regularly confirmed by PCR using Wolb_F and Wolb_R3 primers [72]. Genotyping was confirmed by PCR and Sanger sequencing using the primers in S4 Table.

## CI measurement assays

CI was measured using hatch rate assays. To control for the paternal grandmother age effect on CI [39], virgin *nos*-GAL4:VP16 females were collected for the first 3 days of emergence and aged 9–11 days before crossing to nonvirgin UAS transgenic males. Collections for maternal and paternal lineages were separated by a 7-day period. Individual male and female mating occurred in 8-oz *Drosophila* stock bottles with a grape juice agar plate smeared with yeast and secured to the opening of each bottle with tape. Only the first emerging and youngest males were used to control for the younger brother effect and age effects on CI [40,41]. Grape-juice agar plates were made by first autoclaving a mixture of 12.5 g of agar and 350 ml of de-ionized water. 0.25 g tegosept (methyl 4-hyrdoxybenzoate) dissolved in 10 ml of ethanol and 150 ml of Welch's grape-juice was then added to the autoclaved agar mixture and poured into lids from 35 ×10-mm culture dishes (CytoOne).The flies and bottles were incubated at 25˚C for approximately 16 hours, at which time the grape plates were removed from the bottles and replaced with fresh plates and stored for an additional 24 hours. After this, the initial number of embryos on each plate were counted. The plates were incubated at 25˚C and after 30 hours, the number of unhatched embryos was counted. The percentage of embryos that hatched was calculated by dividing the number of hatched embryos by the total number of embryos and multiplying by 100. Plates with fewer than 25 embryos were excluded from analysis as previously described [23,24]. The specific genotypes of all flies are shown in the Supplementary Data associated with each figure.

## Selection analysis

Selection analysis of CifB was conducted using a sliding window analysis of $K_a$ and $K_s$ (SWAKK). The SWAKK 2.1 webserver first generated a 1196 codon alignment (with gaps) between *cifB*$_{wMel}$ and *cifB*$_{wHa}$ using ClustalW2 and then calculated the Ka/Ks ratio across the gene alignment using a sliding window of 25 codons and a jump size of 1 codon. The *cifA* SWAKK analysis was previously published and is based on the same parameters described above for *cifB* [24].

## Predicting mutational impact on protein structure

The effect of mutations on protein structure was evaluated with the I-TASSER protein prediction tool [42]. I-TASSER generated protein tertiary structure predictions for Cif proteins and their mutants using the on-line server with default settings. Structures are build based on the top ten hits generated by querying the PDB. Hits were provided Z-scores that characterize the similarity to the query sequence. Higher Z-scores represent more confident matches. Z-scores are reported in the source data provided with this manuscript and PDB hits are detailed in S2 Table. C-scores and TM-scores were generated for each tertiary structure. C-scores range from -5 to 2 where 2 is the highest confidence. TM-scores range from 0–1 where 1 is the highest confidence.

## Statistical analysis

All statistical analyses for hatch rates were conducted in GraphPad Prism 8. Hatch rate statistical comparisons were made using Kruskal-Wallis followed by a Dunn's multiple comparison test. All p-values from statistical comparisons are provided in S1 Table. Figure aesthetics were edited using Affinity Designer.

## Supporting information

**S1 Fig. Summary of I-TASSER structural predictions for CifA, CifB, and their mutants.**
(A, C) Venn-diagrams showing the number of PDB hits shared between wild-type and mutant

(A) CifA and (C) CifB proteins. (B, D) I-TASSER used these PDB hits to generate structural predictions for (C) CifA and (D) CifB. TM-scores range from 0–1 where 1 is the highest confidence. RMSD scores are from pairwise alignments of mutant proteins with the wild-type in PyMol. Higher RMSD scores represent more distance between the superimposed proteins. Mutated sites in the tertiary structure are indicated with a white arrow. Domain annotations were based on previous sequence analyses [27]. Details regarding the PDB hits are reported in S2 Table.
(TIF)

**S1 Table. P-values associated with all statistical comparisons made in main and extended data hatch rate and cytology figures.** M = male, F = female,+ = *Wolbachia*-infected, − = *Wolbachia* uninfected.
(XLSX)

**S2 Table. Protein structural prediction software I-TASSER identifies homologous protein domains found in all of our Cif homologs, and those that differ.**
(XLSX)

**S3 Table. Nucleotide and protein sequences used in this study.**
(XLSX)

**S4 Table. Primers used for genotyping and sanger sequencing of *cif* transgenes.**
(XLSX)

**S1 Data. All raw data from main figures, supplemental figures, and replicate data experiments in this study.**
(XLSX)

## Acknowledgments

We thank members of the Bordenstein lab, especially Brittany Leigh, for helpful comments and critiques during the course of this study.

## Author Contributions

**Conceptualization:** J. Dylan Shropshire, Seth R. Bordenstein.

**Formal analysis:** J. Dylan Shropshire, Mahip Kalra.

**Funding acquisition:** J. Dylan Shropshire, Seth R. Bordenstein.

**Investigation:** J. Dylan Shropshire, Mahip Kalra.

**Methodology:** J. Dylan Shropshire, Seth R. Bordenstein.

**Supervision:** J. Dylan Shropshire, Seth R. Bordenstein.

**Validation:** J. Dylan Shropshire.

**Visualization:** J. Dylan Shropshire.

**Writing – original draft:** J. Dylan Shropshire, Seth R. Bordenstein.

**Writing – review & editing:** J. Dylan Shropshire, Mahip Kalra, Seth R. Bordenstein.

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
