## [Decision Letter · Decision Letter 0]

29 Jun 2020

Dear Dr. Shropshire,

Thank you very much for submitting your manuscript "Evolution-guided mutagenesis of the cytoplasmic incompatibility proteins: Identifying 1 CifA’s complex functional repertoire and new essential regions in CifB" for consideration at PLOS Pathogens. As with all papers reviewed by the journal, your manuscript was reviewed by members of the editorial board and by several independent reviewers. The reviewers appreciated the attention to an important topic. Based on the reviews, we are likely to accept this manuscript for publication, providing that you modify the manuscript according to the review recommendations.

Sincerely,

Jason L. Rasgon

Associate Editor

PLOS Pathogens

Kirk Deitsch

Section Editor

PLOS Pathogens

Kasturi Haldar

Editor-in-Chief

PLOS Pathogens

orcid.org/0000-0001-5065-158X

Michael Malim

Editor-in-Chief

PLOS Pathogens

orcid.org/0000-0002-7699-2064

Reviewer Comments (if any, and for reference):

Reviewer's Responses to Questions

**Part I - Summary**

Reviewer #1: This paper elegantly uncovers and beautifully explores the functional domains of the two phage-derived, Wolbachia-encoded proteins CifA and CifB that act as the two key effector proteins for directing cytoplasmic incompatibilities in insects.

Based on their earlier and other´s prior published data they systematically analyzed the protein sequences of CifA and CifB for highly conserved sites across phylogenetically distant Cif genes, site-mutagenized such candidate sites in 4 domains of each wMel gene and expressed them under the control of nanos in transgenic Wolbachia-uninfected flies. By this systematic in vivo approach, the authors have uncovered novel and important aa domains in these phage-derived proteins that affect either the CI, the Rescue or both Wolbachia-expressed phenotypes. Thereby they could demonstrate that in contrast to earlier expectations also the CifA protein, which has been regarded as a pure antitoxin in the TA model, has at least three functional domains that affect CI and two of them also the rescue phenotype. By this, these data further support their “Two-by-One” genetic model of CI.

In addition, they elegantly showed that beside the earlier characterized Ubiquitin-like-specific protease domain earlier proposed by Beckmann as the “enzymatic warhead” for CI three other domains are functionally crucial for CI in vivo.

To sum up, these novel and exciting data are important and pivotal for deepening our understanding in CI and hence this study sets the corner stone for further studies in deciphering the exact biochemical and cell biological mechanisms of these two Type 1 master genes that are driving reproductive parasitism in insects.

Reviewer #2: The manuscript by Shropshire et al. investigates the phenotypic impact of evolution-guided introduced mutations in the two CI associated genes of Wolbachia wMel, CifA and CifB, when transgenically expressed in Drosophila melanogaster. This is the first study that investigates the effect on CI of replacements of conserved amino acid residues across several domains/regions of the CifA and CifB proteins.The results suggest that both proteins are needed for the induction of CI, and that residues along the whole CifB protein, and a large part of CifA are necessary for CI induction, whereas only the N-terminal part of CifA is involved in rescue of CI.

The results support the Two-By-One model, that suggest that both proteins are necessary for CI induction, whereas CifA is responsible for rescue. Additionally, it suggests that several regions/residues of each protein, with different predicted functions or no predicted function at all, has a large impact on the phenotypic expression of CI. Finally, it implies that there are sites in CifA that are only involved in CI induction.

Overall, I think the authors provide good arguments for their conclusions, which are clearly supported by their results.

Given the potential utility of the CI mechanism as a biological control method for insect pests and disease vectors, studies that contribute to further understanding of the mechanism and functions of the Cif proteins is of high significance.

**Part II – Major Issues: Key Experiments Required for Acceptance**

Reviewer #1: (No Response)

Reviewer #2: I don’t have any major issues with the manuscript.

**Part III – Minor Issues: Editorial and Data Presentation Modifications**

Reviewer #1: Line 272: since this part covers CifA and CifB proteins change Fig S1 A to Fig S1A,C and in line 274 Fig S1B to FigS1B,D.

Line 324ff: For a broader audience of readers it would be helpful to introduce the different mechanistic models for CI already in their introduction and phrase out the abbreviations used (i.e., HM and TA).

Reviewer #2: • L362: From looking at figure 2, 34,05% survival is wrong for cifA2 rescue. It looks more like 28% or so to me. Additionally, I think it is a bit confusing that the authors report the percent of embryos that die in the previous sentence (also 34,05%) and the percent that survive in this one. I suggest that the authors use the same measure (i.e. hatch rate as in figure 2), when referring to both mod and resc functions.

• The methods need to be more clearly described overall. Even though many of the methods have been used in other previous publications, I believe that it should be possible to get the main points without reading several additional publications.

o It is possible to infer from the results section which crosses were made, but it is not really described in the Methods section.

o How was the expression of the transgenic genes tested?

o The source data for Fig2 and Fig3 contains repeats, but I can’t find that this is described anywhere in the text.

• I am wondering why the authors chose to do the crosses with the cifA:B dual expression transgenic flies and CifA-types rather than with Wolbachia-infected male when testing the rescue phenotype. Wouldn’t it be relevant to test the CifA variants against the natural Wolbachia infection? Or at least as well as the transgenes, since the expression from Wolbachia might be different than the transgenic expression of CifA and CifB in several ways. I suggest that the authors add at least something small in the discussion about this.

• The illustration in Figure 1B is the same as used in another publication Shropshire and Bordenstein, PLoS Genetics 2019, which could be noted in the legend.

PLOS authors have the option to publish the peer review history of their article (what does this mean?). If published, this will include your full peer review and any attached files.

Reviewer #1: No

Reviewer #2: No
---

## [Editor Report · Decision Letter 1]

9 Jul 2020

Dear Dr. Shropshire,

We are pleased to inform you that your manuscript 'Evolution-guided mutagenesis of the cytoplasmic incompatibility proteins: Identifying CifA’s complex functional repertoire and new essential regions in CifB' has been provisionally accepted for publication in PLOS Pathogens.

Best regards,

Jason L. Rasgon

Associate Editor

PLOS Pathogens

Kirk Deitsch

Section Editor

PLOS Pathogens

Kasturi Haldar

Editor-in-Chief

PLOS Pathogens

orcid.org/0000-0001-5065-158X

Michael Malim

Editor-in-Chief

PLOS Pathogens

orcid.org/0000-0002-7699-2064
---

## [Editor Report · Acceptance letter]

13 Aug 2020

Dear Dr. Shropshire,

We are delighted to inform you that your manuscript, "Evolution-guided mutagenesis of the cytoplasmic incompatibility proteins: Identifying CifA’s complex functional repertoire and new essential regions in CifB," has been formally accepted for publication in PLOS Pathogens.

Best regards,

Kasturi Haldar

Editor-in-Chief

PLOS Pathogens

orcid.org/0000-0001-5065-158X

Michael Malim

Editor-in-Chief

PLOS Pathogens

orcid.org/0000-0002-7699-2064